# Investigation of the Absorption Spectrum of InAs Doping Superlattice Solar Cells

**DOI:** 10.3390/nano14080682

**Published:** 2024-04-16

**Authors:** Ruiqin Peng, Wenkang Su, Zhiguo Yu, Jiamu Cao, Dongwei Jiang, Dongbo Wang, Shujie Jiao

**Affiliations:** 1School of Intelligence Engineering, Shandong Management University, Jinan 250357, China; prq07@sdmu.edu.cn (R.P.); chenzhao@hit.edu.cn (W.S.); yzg@sdmu.edu.cn (Z.Y.); 2School of Astronautics, Harbin Institute of Technology, Harbin 150001, China; 3Key Laboratory of Optoelectronic Materials and Devices, Institute of Semiconductors, Chinese Academy of Sciences, Beijing 100083, China; jdw@semi.ac.cn; 4School of Materials Science and Engineering, Harbin Institute of Technology, Harbin 150001, China; shujiejiao@hit.edu.cn

**Keywords:** InAs, nipi superlattice, subband absorption, mechanism

## Abstract

InAs doping superlattice-based solar cells have great advantages in terms of the ability to generate clean energy in space or harsh environments. In this paper, multi-period InAs doping superlattice solar cells have been prepared.. Current density–voltage measurements were taken both in the dark and light, and the short-circuit current was estimated to be 19.06 mA/cm^2^. Efficiency improvements were achieved with a maximum one sun AM 1.5 G efficiency of 4.14%. Additionally, external quantum efficiency and photoluminescence with different temperature-dependent test results were taken experimentally. The corresponding absorption mechanisms were also investigated.

## 1. Introduction

Solar energy, as a new type of renewable energy, has become an important way to solve the global energy crisis and one of the primary choices for people to obtain clean energy [1]. Usually, there are two ways to utilize solar energy: photothermal conversion and photoelectric conversion. Among them, photoelectric conversion plays a major role in the utilization of solar energy. For photoelectric conversion of the solar cell, photoelectric conversion efficiency is an important technical indicator for judging applications. The efficiency of a solar cell is defined as the percentage of power converted from sunlight to electrical energy under standard test conditions. In 1961, the semi-empirical efforts developed by Walter Shockley and Hans Quesisser detailed the balance model to calculate the efficiency of solar cells, which depicted that a crystalline Si solar cell has the maximum theoretical efficiency of ~30% [2]. In order to obtain high conversion efficiency and low-cost solar cells, some alternative designs have been introduced, including the use of low-cost substrates, novel material systems, or the inclusion of nanostructures, such as dye-sensitized cells [3,4,5,6], perovskite cells [7,8], group III-V multi-junction cells [9], quantum wells [10], and superlattice cells [11]. Dye-sensitized and perovskite cells have been proposed due to their high conversion efficiency and low-cost applications.

However, the difficulties in fabrication and long-term durability are still the biggest obstacles for practical demands. By contrast, group III-V semiconductor thin film or multi-heterojunction solar cells are more mature technologies, especially for quantum well and superlattice structures. Since 1970, when Esaki proposed to develop artificial semiconductors [12], n-type/intrinsic/p-type/intrinsic (nipi) superlattice structures have gained great attention in solar cell production. In the initial solar cell application, nipi is a doping superlattice (DSL) formed by repeating intrinsically doped epitaxial GaAs layers, which are stacked vertically in a parallel connected multi-period solar cell [13]. So far, this format has been widely applied and has found great success in detectors, lasers, and other photonic structures [14,15]. On the other hand, in solar cell structures, III–V multijunction tandem solar cells have been developed for application in space satellites because of their superior radiation tolerance. For applications in space, the solar cells work in a very harsh environment, which features, for example, temperature variations ranging hundreds of degrees, electromagnetic interference, high-energy cosmic radiation particles, etc. Traditional solar absorb materials and device structure are not fit for these applications. So, it is widely believed that a p-i-n-type device design can effectively extend the carrier lifetimes as well as improve radiation tolerance and the formation of subband gap energy states [16,17]. These proposed advantages are beneficial for reducing thermalization and transmission losses, especially in an external harsh environment. Due to the short diffusion length requirements of the absorbing layer, normally, the use of multiple repeated periods of thin nipi layers is recognized as an alternative way to absorb the incoming light; moreover, it is also recognized as a way of minimizing loss in carrier collection. This means smaller losses in a short-circuit current, even with high levels of radiation, so it is ideal for space applications. The nipi superlattice structures based on GaAs have been studied more widely than other materials systems. The absorption edge of the traditional GaAs solar cell is limited to 0.89 μm, and generally, InAs is introduced to expand the light absorption and spectral response (due to the absorption spectrum expanding to infrared region and improving subband collection). However, it should be noted that its conversion efficiency is low, and the device structure leads to higher costs. With the above in mind, the goal is to propose improvement plans based on experimental results in order to optimize the performance of the InAs solar cell. 

In this work, we focus on a new role of the InAs nipi superlattice layer as the absorption unit in solar cell structures, with the intended aim for applications in space at very low temperatures. Meanwhile, the introduction of an InAs absorption layer is hoped to expand the whole absorption edge in order to be comparable with conventional GaAs solar cells or InAs quantum-dot-embedded GaAs solar cells. We also analyze and discuss the spectrum from external quantum efficiency (EQE) and photoluminescence (PL) results of the InAs nipi superlattice, whose corresponding absorption mechanism is investigated. 

## 2. Materials and Methods

The samples were prepared by Riber C21T solid source molecular beam epitaxy (MBE) system (Bezons, France). Figure 1a illustrates the growth process. The structures were grown on GaAs substrates, first a 500 nm thick p-type GaAs buffer and a 200 nm thick p-type InAs contact layer, followed by the nipi InAs superlattice-doped active region and capped with a 100 nm thick n-type InAs layer. Each repeating nipi doping region in the superlattice design was 50 nm thick. The n- or p-type layers were doped with Si or Be, respectively, with doping carrier concentration of 1 × 10^18^ cm^−3^. The light and dark current–voltage curves were measured every 48 h or 12 cycles under the solar simulator system. The light source of the solar simulator was an OSRAM 1000 W short-arc xenon lamp (Munich, Germany), which can simulate the spectrum of AM 1.5 G standard irradiance.

The layers showed a sinusoidal-like band alignment due to the alternating n- and p-type layers for this structure (as shown in Figure 1b). It results in high carrier extraction efficiency and minimal sensitivity to minority carrier diffusion length, providing a good fit for high-temperature or high-radiation applications. Furthermore, the device was produced by employing traditional photolithography followed by etching. The structures were etched down to the p-type GaAs buffer layer. The n-type and p-type metallization of Ti/Pt/Au contacts were deposited followed by a metal lift-off. All samples were passivated with hydrochloric acid-isopropanol solution (HCl-IPA), which eliminates any remains from etching and preserves the atomic smoothness of the surface.

## 3. Results and Discussion

Current density–voltage curves measured under dark and bright conditions are shown in Figure 2. The results indicate that the number of periods in the nipi superlattice has a significant influence on the dark current. A suitable number of superlattice cycles would help to achieve a balance between a lower dark current and higher efficiency. As seen in Figure 2a, as the number of periods increases, the dark current attains a relatively low value. When the number of periods is 75, the corresponding dark current is higher than that for other values, and this increase in dark current is mainly due to the multiple parallel connected junctions (as shown in Figure 1a).

Generally, the repeated connected junctions all contribute to the dark current; however, the crystalline quality also improves with increasing superlattice periods to some extent, which could be helpful in reducing the dark current. When the number of periods is smaller than 50, the relatively low dark current can be attributed to the improved uniformity and stain-free quality of the epitaxial layers. As the number of periods is high (up to 50), the dark current strongly increases, which can be attributed to the increase in lattice stress and defects with the increase in repeated period thickness.

In order to present the device performance clearly, the relevant dark current was extracted by fitting the curves with the dual diode dark current equation [18]:(1)$$J_{dark}\left(V\right)=J_{01}\left(e^{\frac{qV}{kT}}-1\right)+J_{02}\left(e^{\frac{qV}{2kT}}-1\right)$$
where *q* is the electron charge, *k* is the Boltzmann constant, and *T* is the temperature. *J*_01_ and *J*_02_ are the dark current saturation currents. The factor *J*_01_ is related to the Shockley–Reed–Hall (SRH) generation/recombination rate—this process occurs more rapidly in the depletion region. The *J*_02_ term is described in detail by the following proportionality:(2)$$J_{02}\propto \frac{qv_{th}W}{2}\sigma N_{t}$$
where *W* is the depletion width, *ν_th_* is the carrier thermal velocity, *σ* is the carrier capture cross-section, and the *N_t_* is the number of traps.

From the preceding two equations, the dark currents *J*_01_ and *J*_02_ can be calculated as well as the corresponding short-circuit current. The open-circuit voltage is also shown in Figure 2. The samples were numbered 1 to 4 (with number of periods N1 to N4 given in Figure 2a), and their corresponding efficiencies reached 2.089%, 2.548%, 4.139%, and 3.410%, respectively. As can be concluded from the above, these solar cells all have higher dark saturation currents and the ideality factors are higher than expected, partly due to a shunt at the superlattice interface. Additionally, there is an increase in cumulative depletion region thickness, which also contributes to the increased dark current component [19]. Thus, there is also an efficiency improvement in sample N3 over prior results from 3.42% [12].

Compared with a typical GaAs single junction VOC of 1.05 V, the open-circuit voltage (~0.4 V) is significantly reduced. Such a loss in VOC is mainly due to the nipi design, which can be primarily explained by a bigger dark current than that of the GaAs single junction due to there being more parallel junctions in nipi superlattices, or put another way, it is believed to be due to the epitaxial layers of the junctions forming interface traps. From these results, it can be speculated that the voltage and efficiency will increase significantly with decreasing interface traps and increasing doping concentration, and this idea can be investigated in the future.

In order to identify the processes, EQE and PL results are shown in Figure 3. In Figure 3a, the peaks were extracted from EQE measurements where the main emission peak can cover the entire visible region, and the subband peak was at 938 nm. The absorption edge can be extended to 1050 nm for EQE, which has a broadened absorption range compared with the reference GaAs p-i-n solar cell, demonstrating substantial enhancement of the EQE reaching over 30% in the wide range of 900–1000 nm. The results show that it is feasible to expand the range of spectral response with doping nipi superlattice. Additionally, we investigated the effects of temperature on optical properties. The temperature-dependent PL emission spectra of samples are plotted in Figure 3b. It can be expected for the PL peak position to shift monotonously to lower energy with increasing temperature due to optical gap shrinkage. This can be analyzed using the Varshni formula [20], given as follows:(3)$$E_{g}\left(T\right)=E_{0}-\frac{\alpha T^{2}}{\beta +T}$$
where *E*_0_ is the band gap at 0 K and *T* is the temperature. *α* (eV/K) and *β* (K) are Varshni coefficients. The fitting parameter *α* was found to be 0.315 meV/K. The fitting parameter *β* is equal to 101.7 K. Compared with the reported energy bandgap of the bulk InAs, GaAs crystals, and other structures [21,22], it has been revealed that in this structure the fitting parameters in the temperature range 4–300 K are very close to their values for InAs quantum dot structures [19], differing slightly from the values in InAs. It is still, however, in alignment with InAs-like bandgap shrinkage, as shown by the Varshiapproximation. Furthermore, from these experimental results, we can see that there is some mismatch between PL and EQE peak locations, due to Stokes shift [23].

The occurrence of this subband gap peak in EQE can be explained by the combination of two different effects. The first effect is the confined states within the wells, which is originated from minibands and was formed in the forbidden gap of the semiconductor. This mechanism has the strong effect near the valence or conduction band edge, which requires the least energy. It is the first absorption mechanism, which is depicted in a confined state, as shown in Figure 4.

The second effect contributing to subband absorption can be recognized as the Franz–Keldysh effect. Due to high electric fields through the SLs, the effect normally describes the overlap of electron and hole wavefunctions within the forbidden gap. The existence of an electric field-dependent absorption is explained as the photo-assisted tunneling of electrons from the valence to the conduction band. The electric field is a function of position in the growth axis z, which is described by the following equations [18,24]:(4)$$\varepsilon \left(z\right)=\frac{1}{\epsilon }\int \rho \left(z\right)dz=\left\{\begin{array}{l} -\frac{qN_{A}}{\epsilon }\left(\frac{d_{p}^{0}}{2}-\left|z\right|\right)\;\;\;\;\;\;\;\;\;\mathrm{if}\;\frac{d_{p}^{0}}{2}<\left|z\right|<\frac{d_{p}}{2}\\ \frac{qN_{A}}{\epsilon }\frac{d_{p}-d_{p}^{0}}{2}\;\;\;\;\;\;\;\;\;\;\;\;\;\;\;\;\;\;\mathrm{if}\;\frac{d_{p}}{2}<\left|z\right|<\frac{{d-d}_{n}}{2}\\ \frac{qN_{D}}{\epsilon }\left(\left|z\right|-\frac{d-d_{n}^{0}}{2}\right)\;\mathrm{if}\;\frac{{d-d}_{n}}{2}<\left|z\right|<\frac{d-d_{n}^{0}}{2} \end{array}\right.$$
where *d_n_*^0^ and *d_p_*^0^ are the n- and p-type quasi-neutral region thickness, *d* is the entire nipi period thickness, and *ϵ* is the relative permittivity. *N_D_* is the donor concentration, *N_A_* is the acceptor concentration, and *d_n_* and *d_p_* are the n-type and p-type thickness, respectively. The electric field within each of the doped and intrinsic layers results in rapid carrier collection with minimal recombination, which is from the calculation in Equation (3). As shown in the reported paper, if each layer thickness is about 50 nm with a doping concentration of 1 × 10^18^ cm^−3^, the electric field would reach a maximum value exceeding 3.6 × 10^5^ V/cm [12]; this effect leads to a significant overall contribution to absorption.

As previous works [25] have showed, such an interband optical absorption should include the effects of the Coulomb interaction of the electron and hole, and it results in exciton resonances. Thus, it is known that the excitonic resonances in semiconductors are strongly weakened by high electric fields. Therefore, the SL design mainly determines the subband gap absorption that each effect contributes. To some extent it is also clear that the total subband absorption cannot be entirely explained by one of these two effects, which means that the Franz–Keldysh effect is not the only absorption mechanism. Based on the above analysis, the subband absorption in EQE can be recognized as a combination of these two different effects.

In the range above 900 nm, compared with traditional GaAs solar cells, this subband absorption contribution is remarkable. Theoretical analysis of the subband collection absorption in a GaAs DSL has been demonstrated elsewhere, and we expect that the subband absorption of InAs nipi doping superlattice would be similar, with even more subband absorption in the infrared region based on inherent material properties. Therefore, in order to achieve an increase in subband absorption, it is necessary to introduce the thin nipi layers with an appropriate number of cycles, and increase the doping levels to maximize the built-in field at each junction. These measures will be beneficial for further improving device performance and achieving efficiency of InAs nipi doping superlattice solar cell solar cells.

## 4. Conclusions

In summary, the influence of the number of periods in InAs nipi doping superlattice solar cells was investigated. The samples were measured under dark and light conditions to obtain current density curves, and the corresponding electrical properties were presented. The short-circuit current was estimated to be 19.06 mA/cm^2^, and the maximum one sun AM 1.5 G efficiency achieved was 4.14%. The EQE and PL test results were taken experimentally and the subband absorption mechanisms were also clearly investigated. The results provide guidance on how to achieve the desired efficiency and increase subband collection absorption in InAs nipi doping superlattice solar cells.

## Figures and Tables

**Figure 1 nanomaterials-14-00682-f001:**
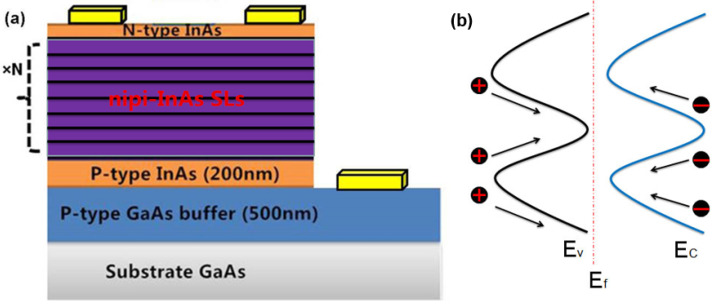
(**a**) Depiction of nipi cross-section and (**b**) the corresponding band diagram of the nipi active region.

**Figure 2 nanomaterials-14-00682-f002:**
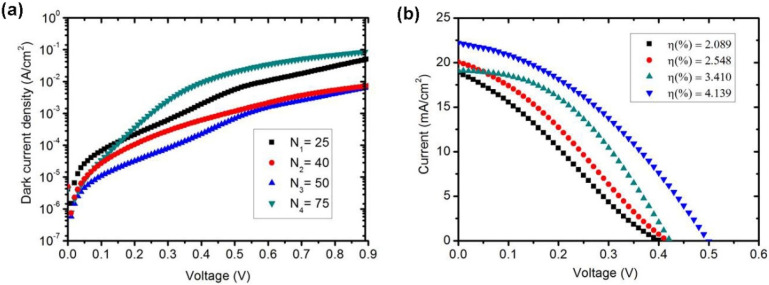
(**a**) Current–voltage characteristic of solar cells with different numbers of nipi periods under 1 sun AM 1.5 illumination. (**b**) the open-circuit voltage under different efficiency.

**Figure 3 nanomaterials-14-00682-f003:**
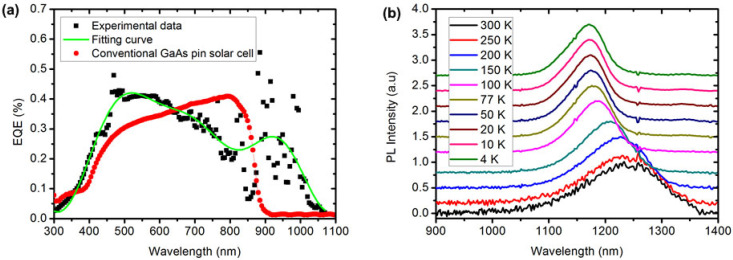
(**a**) The differentials of the external quantum efficiency between the InAs nipi solar cell (sample N3) and reference GaAs p-i-n solar cell. (**b**) PL spectrum measured at different temperatures for sample N3, shown to slight redshift with InAs-like bandgap shrinkage tendency.

**Figure 4 nanomaterials-14-00682-f004:**
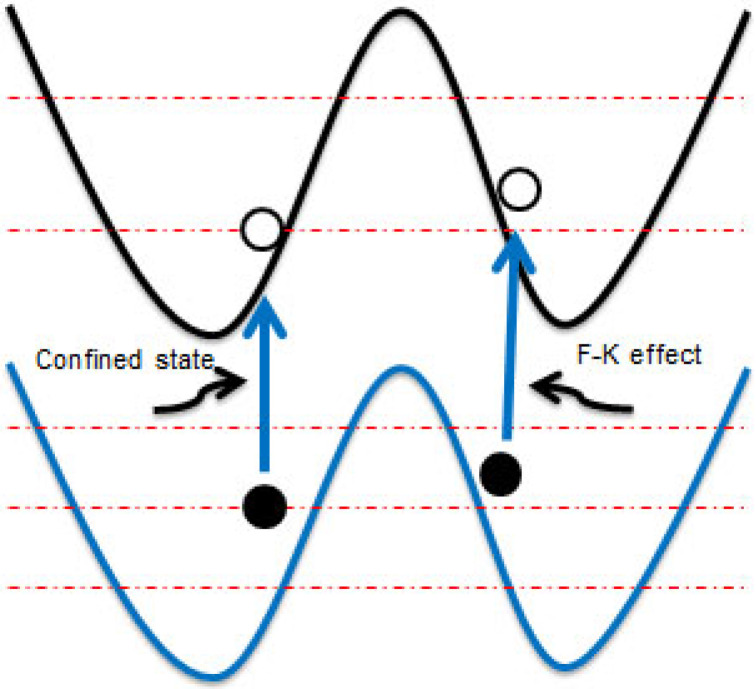
Band diagram depicting the two main mechanisms of subband absorption, namely collection through the confined state and the Franz–Keldysh effect. The red dotted in the figure means the different energy band.

## Data Availability

Data are contained within the article.

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
