# Peer review of "Investigation of the Absorption Spectrum of InAs Doping Superlattice Solar Cells"

_nanomaterials, 2024, doi:10.3390/nano14080682_

Round 1

Reviewer 1 Report

Comments and Suggestions for Authors

The article covers a correct study of multi-period InAs nipi doping superlattice solar cells. The work partially contains original elements as well as review of other works carried out so far in this field.. In my opinion, it is well organized and written and supported by appropriate references. The methodology was properly chosen. The results are clearly described and well discussed. The article is of sufficient importance for this field so, it  may be published in “Nanomaterials” journal after minor revision.

I have a few comments that I would definitely suggest you consider. The article is strikingly short, but publishing it in this form is a bit risky because one has to be an expert in all fields and all techniques he is referring to, but now all techniques are not fully presented. For example, the conditions of the external quantum efficiency and photoluminescence with different temperature measurements should be somewhat detailed. Likewise, it is not entirely satisfactory the description of the current density-voltage measurements under dark and bright conditions. There are formal errors in lines 44,68, 104-114, 175-6; the notation [1] of the equation after line 139 is obviously a typo.

Author Response

Dear Editor and Reviewers,

Thank you for your letter and for the reviewer’s comments concerning our

manuscript entitled “ Investigation of absorption spectrum of InAs doping superlattice solar cell”. (nanomaterials-20240129). Those comments are all valuable and very helpful for revising and improving our paper. We have carefully made corrections. The revised portions are marked in red in the manuscript. The corrections in the manuscript and the response to the reviewer’s comments are as follows: 

Comment 1: The article covers a correct study of multi-period InAs nipi doping superlattice solar cells. The work partially contains original elements as well as review of other works carried out so far in this field. In my opinion, it is well organized and written and supported by appropriate references. The methodology was properly chosen. The results are clearly described and well discussed. The article is of sufficient importance for this field so, it  may be published in “Nanomaterials” journal after minor revision.

I have a few comments that I would definitely suggest you consider. The article is strikingly short, but publishing it in this form is a bit risky because one has to be an expert in all fields and all techniques he is referring to, but now all techniques are not fully presented. For example, the conditions of the external quantum efficiency and photoluminescence with different temperature measurements should be somewhat detailed. Likewise, it is not entirely satisfactory the description of the current density-voltage measurements under dark and bright conditions. There are formal errors in lines 44,68, 104-114, 175-6; the notation [1] of the equation after line 139 is obviously a typo.

Response: Thanks for your comments. As your suggestion, the word number of our article is not fit for the research paper, we changed the article type for publishing in short communication type, which is well covered. Otherwise, we list the conditons of the EQE and PL measurements. At last, we checked and revised some formal errors and the details have been rewritten according to your useful advice, and the corresponding revision has been done and marked in red.

Comment 2: The paper presents an investigation into the absorption spectrum of InAs doped nipi superlattice solar cells, emphasizing their potential for space applications due to their efficiency and durability in extreme conditions. It explores the manufacturing process, utilizing molecular beam epitaxy, and examines the solar cells' performance under various conditions. The study highlights how InAs layers enhance light absorption into the infrared spectrum, improving efficiency and sub-band collection. Results indicate a notable efficiency of 4.14% and a short-circuit current of 19.06 mA/cm². The research further delves into the mechanisms behind sub-band absorption, attributed to both confined states within wells and the Franz-Keldysh effect.

Shortcomings of the study include

1) a comparison with other high-efficiency solar cell technologies,

2) an analysis or expectation of the cost implications and manufacturing challenges that might limit their practical deployment.

3) Further, while the paper discusses the theoretical basis for efficiency improvements, it could benefit from a more comprehensive analysis of long-term stability and performance.

Response: Thanks for your comments. As you said, in the first part of the article, we compared and analyzed several mainstream solar cell structures, and proposed a solar cell based on superlattice, whose special structure can adapt to complex environments, like space application. But I have to admit that its conversion efficiency is low. Based on your suggestion, we have added a description of the problems with the solar cells of this structure and marked them in red.

Comment 3: The article requires thorough proofreading, both linguistic and writing. Literature should be cited uniformly (with a break in relation to the text). For example, the sentence on line 33 must begin with a capital letter. Many language mistakes should be avoided by using the help of a native speaker. The number of references should be increased in the Introduction. Reference to different types of solar cells should be better supported in the bibliography (e.g., perovskite cells cf. Materials 2022, 15, 2254. https://doi.org/10.3390/ma15062254 and references therein, the same applies to conventional p-n junction cells such as Si or CIGS based cells). It is worth mentioning in the Introduction the variety of methods for increasing photovoltaic efficiency, including a more detailed comment on the efficiency limit of the solar cell - the so-called the Shockley-Queisser efficiency limit and methods of exceeding it, e.g., tandem solutions, special preparation of cell surfaces, use of quantum dots or plasmonic elements (in the case of the latter in Si, CIGS and in thin-film cells, see the overview of experiments and detailed microscopic theory in "Quantum Nano -Plasmonics”, Cambridge UP 2020). The potential of particular methods for increasing cell efficiency should be compared, emphasizing the difference depending on the type of cell operation. It is worth mentioning the basic difference between cells with a p-n junctions (including the cell discussed in the article) and chemical cells (including  hybrid perovskite cells without p-n junctions) (see Nano Energy 75 (2020) 104751). Based on such a short but more comprehensive review and comparison, the considered "nipi" solution should be initially described, pointing to the physical phenomenon that causes the associated growth of the III-V thin film cell efficiency. What does mean "n-i-p"  "p-i-n" in lines 34, 42, 58, 59. What does mean "cm-3" in line 68. The paragraph Materials and Methods should be more precisely developed. The quality of graphics in Fig. 1 should be enhanced.

Response: Thanks for your comments. We've taken your suggestion and made changes accordingly.

Comment 4: I would like to see more information about the experimental details regarding efficiency and EQE. Also some more references regarding the role of InAs nipi and comparison with GaAs nipi efficiency from the literature.

Response: Thanks for your comments. We have listed the details in the paper. Both light and dark current-voltage curves were measured on each cell every 48 h or 12 cycles under the solar simulator system in order to have a good tracking of the degradation, and the other electrical parameters of short-circuit current density (Jsc), open-circuit voltage (Voc), and photoelectric conversion efficiency (Eff) were also obtained. In order to avoid the effect of series resistance, the four-point probe method was used. The light source of the solar simulator is OSRAM 1000W short-arc xenon lamp, which can simulate the spectrum of AM1.5G standard irradiance.

We try our best to improve the manuscript and correct some small errors in the manuscript. We appreciate the editor and reviewer’s warm work suggestion and hope that the correction would meet with the requirement for publication.

Once again, thank you very much for your comments and suggestions!

Yours sincerely,

Ruiqin Peng, Dongbo Wang

Corresponding author: Dongbo Wang

Reviewer 2 Report

Comments and Suggestions for Authors

The paper presents an investigation into the absorption spectrum of InAs doped nipi superlattice solar cells, emphasizing their potential for space applications due to their efficiency and durability in extreme conditions. It explores the manufacturing process, utilizing molecular beam epitaxy, and examines the solar cells' performance under various conditions. The study highlights how InAs layers enhance light absorption into the infrared spectrum, improving efficiency and sub-band collection. Results indicate a notable efficiency of 4.14% and a short-circuit current of 19.06 mA/cm². The research further delves into the mechanisms behind sub-band absorption, attributed to both confined states within wells and the Franz-Keldysh effect.

Shortcomings of the study include

1) a comparison with other high-efficiency solar cell technologies,

2) an analysis or expectation of the cost implications and manufacturing challenges that might limit their practical deployment.
3) Further, while the paper discusses the theoretical basis for efficiency improvements, it could benefit from a more comprehensive analysis of long-term stability and performance.

Comments on the Quality of English Language

minor revision

Author Response

(The authors gave the same response as above.)

Reviewer 3 Report

Comments and Suggestions for Authors

The article requires thorough proofreading, both linguistic and writing. Literature should be cited uniformly (with a break in relation to the text). For example, the sentence on line 33 must begin with a capital letter. Many language mistakes should be avoided by using the help of a native speaker. The number of references should be increased in the Introduction. Reference to different types of solar cells should be better supported in the bibliography (e.g., perovskite cells cf. Materials 2022, 15, 2254. https://doi.org/10.3390/ma15062254 and references therein, the same applies to conventional p-n junction cells such as Si or CIGS based cells). It is worth mentioning in the Introduction the variety of methods for increasing photovoltaic efficiency, including a more detailed comment on the efficiency limit of the solar cell - the so-called the Shockley-Queisser efficiency limit and methods of exceeding it, e.g., tandem solutions, special preparation of cell surfaces, use of quantum dots or plasmonic elements (in the case of the latter in Si, CIGS and in thin-film cells, see the overview of experiments and detailed microscopic theory in "Quantum Nano -Plasmonics”, Cambridge UP 2020). The potential of particular methods for increasing cell efficiency should be compared, emphasizing the difference depending on the type of cell operation. It is worth mentioning the basic difference between cells with a p-n junctions (including the cell discussed in the article) and chemical cells (including  hybrid perovskite cells without p-n junctions) (see Nano Energy 75 (2020) 104751). Based on such a short but more comprehensive review and comparison, the considered "nipi" solution should be initially described, pointing to the physical phenomenon that causes the associated growth of the III-V thin film cell efficiency. What does mean "n-i-p"  "p-i-n" in lines 34, 42, 58, 59. What does mean "cm-3" in line 68. The paragraph Materials and Methods should be more precisely developed. The quality of graphics in Fig. 1 should be enhanced (the figure caption must be more informative with the explanation of all used symbols in the graphic). Conclusions must be more precisely formulated. In particular, the last sentence in Conclusions is not clear enough. 

The submission needs a revision.

Comments on the Quality of English Language

thorough language proofreading with the help of a native speaker is required

Author Response

(The authors gave the same response as above.)

Reviewer 4 Report

Comments and Suggestions for Authors

I would like to see more information about the experimental details regarding efficiency and EQE. Also some more references regarding the role of InAs nipi and comparison with GaAs nipi efficiency from the literature.

Author Response

(The authors gave the same response as above.)

Round 2

Reviewer 3 Report

Comments and Suggestions for Authors

The Authors have made some revision of the submission. However, certain corrections are avoided. For example, there is still no explanation for the acronym p-i-n. Language verification is necessary. (on line 184 a pause before [11] is necessary, avoid line 214). I recommend publication provided that the text is revised again in accordance with the lines mentioned in the first round of evaluation (some are not met, e.g., the mention of the Shocley-Queisser limit for a cell efficiency).

Comments on the Quality of English Language

Extensive lingustic verification is required

Author Response

Dear Editor and Reviewers,

Thank you for your letter and for the reviewer’s comments concerning our

manuscript entitled “ Investigation of absorption spectrum of InAs doping superlattice solar cell”. (nanomaterials-2864775). Those comments are all valuable and very helpful for revising and improving our paper. We have carefully made corrections. The revised portions are marked in red in the manuscript. The corrections in the manuscript and the response to the reviewer’s comments are as follows: 

Comment 1: The Authors have made some revision of the submission. However, certain corrections are avoided. For example, there is still no explanation for the acronym p-i-n. Language verification is necessary. (on line 184 a pause before [11] is necessary, avoid line 214). I recommend publication provided that the text is revised again in accordance with the lines mentioned in the first round of evaluation (some are not met, e.g., the mention of the Shocley-Queisser limit for a cell efficiency).

Response: Thanks for your comments. As your suggestion, we checked and revised some formal errors and the details have been rewritten according to your useful advice, and the corresponding revision has been done and marked in red.

We try our best to improve the manuscript and correct some small errors in the manuscript. We appreciate the editor and reviewer’s warm work suggestion and hope that the correction would meet with the requirement for publication.

Once again, thank you very much for your comments and suggestions!

Yours sincerely,

Ruiqin Peng, Dongbo Wang

Corresponding author: Dongbo Wang
